# Nb/Ta systematics in arc magma differentiation and the role of arclogites in continent formation

Ming Tang[1], Cin-Ty A. Lee[1], Kang Chen[2], Monica Erdman[1], Gelu Costin[1] & Hehe Jiang[1]

The surfaces of rocky planets are mostly covered by basaltic crust, but Earth is unique in that it also has extensive regions of felsic crust, manifested in the form of continents. Exactly how felsic crust forms when basaltic magmas are the dominant products of melting the mantles of rocky planets is unclear. A fundamental part of the debate is centered on the low Nb/Ta of Earth's continental crust (11–13) compared to basalts (15–16). Here, we show that during arc magma differentiation, the extent of Nb/Ta fractionation varies with crustal thickness with the lowest Nb/Ta seen in continental arc magmas. Deep arc cumulates (arclogites) are found to have high Nb/Ta (average ~19) due to the presence of high Nb/Ta magmatic rutiles. We show that the crustal thickness control of Nb/Ta can be explained by rutile saturation being favored at higher pressures. Deep-seated magmatic differentiation, such as in continental arcs and other magmatic orogens, is thus necessary for making continents.

[1] Department of Earth, Environmental and Planetary Sciences, Rice University, Houston, TX 77005, USA. [2] State Key Laboratory of Geological Processes and Mineral Resources, China University of Geosciences, 430074 Wuhan, China. Correspondence and requests for materials should be addressed to M.T. (email: tangmyes@gmail.com) or to C.-T.L. (email: ctlee@rice.edu)

On average, continental crust and subduction zone magmas have remarkably similar major and trace element compositions, suggesting an intrinsic relationship between continent growth and arc magmatism[1,2]. In particular, both evolved arc magmas and continental crust exhibit low Nb/Ta ratios[1,3,4]. Exactly when arc magmas or the continental crust develop low Nb/Ta ratios is unclear. Are the low Nb/Ta ratios inherited from the source, such as through slab melting or through differential transport in hydrous fluids[5], or are they a feature imparted during intracrustal differentiation, appearing only later in the petrogenetic evolution of arc magmas?

We first approach this problem from a global survey of arc magmas. We show that Nb and Ta can fractionate during arc magma differentiation, but the fractionation primarily happens in arcs built on thickened crust—magmatic orogens. We then turn to the Nb/Ta systematics of deep arc cumulates, represented by garnet pyroxenite xenoliths from Arizona, USA, which crystallized at 45–80 km[6]. We find that those rutile-bearing cumulates possess high Nb/Ta ratios that are complementary to the low Nb/Ta continental crust.

## Results

**The importance of intracrustal differentiation on Nb/Ta fractionation.** Using global arc magma compilations, we find that primitive arc magmas (basalts) have average Nb/Ta identical to that of mid-ocean ridge basalts (MORB) (Fig. 1a, Methods), suggesting that the influence of the slab or mantle on Nb/Ta systematics in arc magmas is small. Only with progressive intracrustal differentiation does Nb/Ta decrease. In mature continental arcs, where the crustal thickness is generally greater than 50 km[7,8], Nb/Ta decreases when $SiO_2$ exceeds 60 wt.%, approaching values as low as ~9. In contrast, the Nb/Ta in island arcs, characterized by thinner crust[7], decreases only after ~70 wt. % $SiO_2$ (Fig. 1a, b). These observations indicate that, in most cases, the low Nb/Ta is not inherited from the source, but instead a feature of intracrustal differentiation.

The importance of intracrustal differentiation is borne out by a possible influence of crustal thickness on Nb/Ta. Nb/Ta shows a strong negative correlation with Dy/Yb for high silica magmas (Fig. 1c). High Dy/Yb ratios in magmas are a unique signatures of garnet crystallization, which is favored during high pressure igneous differentiation[9–14], with high-pressure crystallization itself favored in arcs characterized by thick crust[7,15–17]. These observations indicate that the intracrustal differentiation important for fractionating Nb/Ta is most pronounced in thick arcs. In fact, published models of the Nb/Ta of average continental crust, which are based on weighted averages of upper crustal and lower crustal endmember compositions[1], can only be reproduced by mixing basalts and differentiated magmas from thick continental arcs. Similar mixing calculations applied to island arc differentiation series yield high Nb/Ta at intermediate $SiO_2$ and do not match continental crust model compositions (Fig. 1a).

**Nb/Ta systematics in deep arc cumulates (arclogites).** The low Nb/Ta ratios seen in differentiated magmas from thick magmatic arcs predict that deep arc cumulates, particularly beneath continental arcs, should have the complementary high Nb/Ta. However, because Nb and Ta are extremely incompatible in most of the major silicate minerals, including garnet[18], crystallization of a Nb- and Ta-rich accessory phase that co-precipitates with garnet at high pressure is required. We examined Cretaceous-Paleogene garnet pyroxenite xenoliths from central Arizona[6]. These pyroxenites, also known as arclogites to distinguish them from "eclogites" of an oceanic crust protolith, are thought to represent deep-seated cumulates in a continental arc; they have

major element compositions similar to arc cumulates from the Sierra Nevada and Kohistan arcs[6,15]. These arclogites are dominated by garnet and clinopyroxene, with the more primitive ones characterized by higher clinopyroxene mode (garnet/clinopyroxene down to 0.2), low bulk Fe, and high Mg# (atomic Mg/(Mg + Fe)), and the more evolved ones characterized by high garnet mode (garnet/clinopyroxene > 1), high bulk Fe, low silica, and low Mg#[6]. The more evolved, low Mg# arclogites have been shown to have the necessary compositions to drive the Fe depletion and Si enrichment seen in continental arcs and average continental crust[15,19,20].

Accessory minerals include oxides (rutiles and Fe–Ti oxides) and apatite. In particular, rutiles and Fe–Ti oxides are primarily present in the low Mg# (<0.6) arclogites and typically absent from more primitive arclogites. In the low Mg# arclogites, rutile crystals exist as discrete phases or as inclusions in garnet and clinopyroxene. Consistent with the change in mineralogy is the observation that whole rock $TiO_2$, Nb, and Ta concentrations increase significantly only after Mg# decreases to 0.6 or lower (Fig. 2a–c, Methods). This observation indicates that rutile and/or Fe–Ti oxides saturate in the magma only after sufficient differentiation. Importantly, we find that most of the rutile-bearing low Mg# arclogites have whole rock Nb/Ta ratios greater than average basalt values (Fig. 2d). Sample PR-78-DS is the only low Mg# arclogite that has a low Nb/Ta ratio, but rutiles and Fe–Ti oxides are rare in this sample despite its evolved composition. Excluding PR-78-DS, the average Nb/Ta ratio of all remaining low Mg# samples is 18.8 ± 2.9 (1σ). We do not consider the Nb/Ta systematics of the high Mg# samples (average Nb/Ta = 12.6 ± 5.8, 1σ) because total Nb, Ta, and Ti concentrations are so low (due to absence of rutiles and Fe–Ti oxides), making the high Mg# arclogites insignificant reservoirs of Nb and Ta.

To evaluate whether the rutiles and/or Fe–Ti oxides have high Nb/Ta ratios, we used laser ablation inductively coupled plasma mass spectrometry to perform in situ analyses of the oxides in our arclogite samples. Our analyses confirm that rutiles and Fe–Ti oxides are enriched in Nb (up to 2000 ppm) and Ta (up to 90 ppm). Rutiles ($TiO_2$ > 90 wt.%) tend to have higher Nb/Ta ratios than Fe–Ti oxides (Ti/Fe atomic ratio = 0.6–2.4) (Fig. 3a). Although scattered, most rutile analyses show higher Nb/Ta ratios than arc basalts and MORB (Fig. 3a), suggesting that rutiles are responsible for the high Nb/Ta of low Mg# arclogites. We note, however, that the high Nb/Ta ratios in arclogite rutiles seem inconsistent with most experimental observations, which show $D_{Nb}/D_{Ta} < 1$ between rutile and melt[21–25], but such experiments were done at temperatures >1000 °C, higher than the temperatures at which these rutile-bearing arclogites likely formed (800–1000 °C). Xiong and coworkers[26] recently showed that $D_{Nb}/D_{Ta}$ increases with differentiation and eventually exceeds 1 at temperatures below 1000 °C and water content <10 wt.%. That rutiles and Fe–Ti oxides saturated as magmatic phases only in the low Mg# arclogites, suggests that they crystallized from evolved and cooler magmas; indeed, the low Mg# arclogites are thought to be complementary to andesitic to dacitic magmas and thus crystallized at temperatures of <1000 °C[15].

**Pressure control on rutile solubility.** We now explore why the rutile effect appears to be unique to continental arc differentiation, as marked by the onset of rapid Nb/Ta decline at $SiO_2$ of 60 −65 wt.% and the negative correlation between Nb/Ta and Dy/Yb in high silica arc magmas (Fig. 1). We suggest that this effect may be related to pressure, given that the average igneous differentiation depth, and thus pressure, appears to increase with crustal thickness[7,16,17]. Rutile solubility in silicate melts is a function of temperature, pressure, melt composition, and water

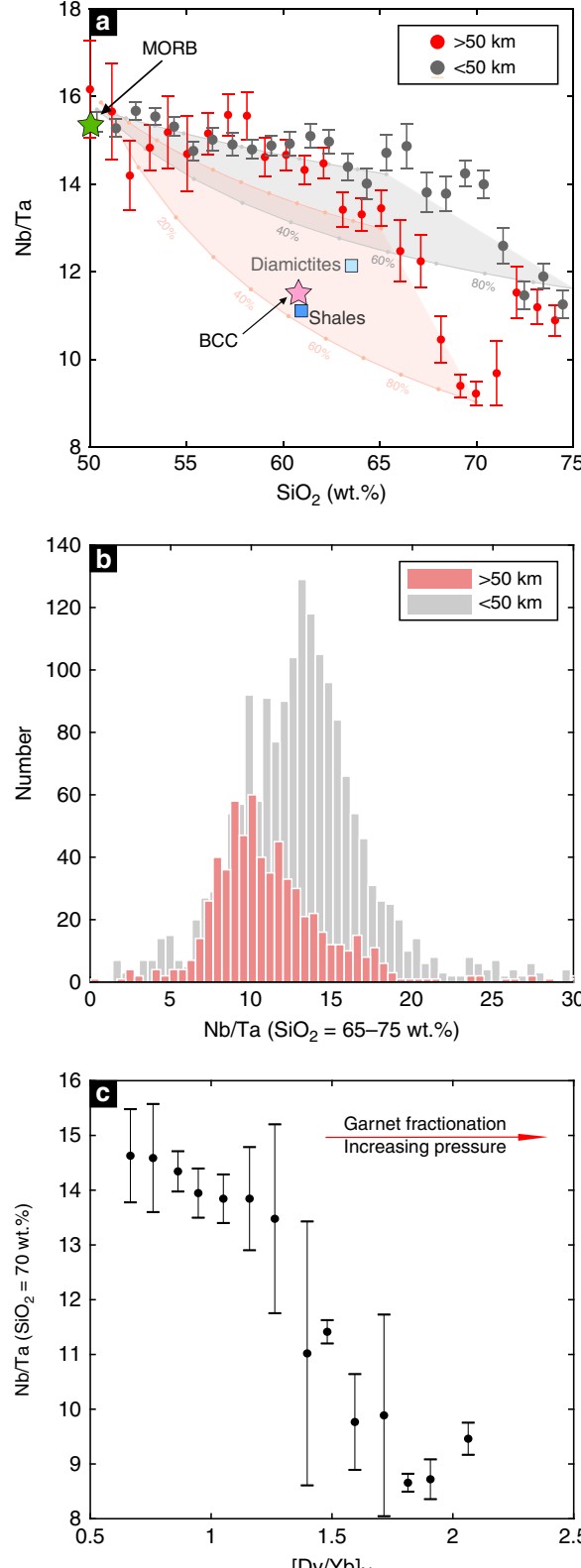

**Fig. 1** Nb-Ta systematics during intracrustal differentiation of arc magmas. **a** Nb/Ta vs. SiO$_2$ for mature continental arcs (crustal thickness >50 km) vs. island arcs and incipient continental arcs (crustal thickness <50 km). Mature continental arc samples are from the central to northern Andean arc. Shown in the plot are mean and two standard error uncertainties as a function of SiO$_2$ (samples are binned in 1 wt.% SiO$_2$ increments). The pale pink and gray shaded areas indicate the mixing fields between basalt (SiO$_2$ = 50 wt.%) and differentiated rocks in arcs with >50 km and <50 km crustal thickness, respectively. The endmember mixing curves are marked at 10% mixing intervals. For thick continental arcs, we used samples with 65 wt.% and 70 wt.% SiO$_2$ as the differentiated endmembers; for arcs built on crust <50 km, we used samples with 65 wt.% and 75 wt.% SiO$_2$ as the differentiated endmembers. The average MORB Nb/Ta (15.4) are from Gale et al.[61]. The bulk continental crust (BCC) value is from Rudnick and Gao[1]. Average post-Archean shale and diamictite compositions[62,63] are also plotted as they are thought to represent the average compositions of insoluble elements in the upper continental crust. **b** Nb/Ta ratios of arc rocks with 65−75 wt.% SiO$_2$ show a systematic difference between mature continental arcs (crustal thickness > 50 km) vs. island arcs and incipient continental arcs (crustal thickness < 50 km). **c** Nb/Ta ratio of differentiated arc rocks at SiO$_2$ = 70 ± 1 wt.% negatively correlates with Dy/Yb, an index of garnet fractionation and differentiation pressure. Nb/Ta ratios are plotted as averages binned by chondrite[64] normalized Dy/Yb binned into 0.1 increments of Dy/Yb. Error bars in **a** and **c** are two standard errors (2 se). For each SiO$_2$ and Dy/Yb bins, we removed 10% of the samples with the highest values and 10% with the lowest values. Data are from GeoRoc[65]

calibrated rutile solubility model[29] and average arc magma compositions as a function of SiO$_2$ content, we find that, in andesitic to dacitic magmas of 950–900 °C and typical TiO$_2$ contents of 0.8–0.6 wt.% (Fig. 3b), rutile saturates only at pressures greater than 1.2–1.5 GPa, corresponding to depths greater than 40–50 km (Methods). At lower pressures, rutile saturation can only occur in rhyolitic and colder magmas. The pressure dependence of rutile solubility in silicate melts thus explains the negative correlation between Nb/Ta and Dy/Yb in evolved arc magmas (Fig. 1c).

## Discussion

Owing to the abundance of garnet, arclogites are denser than the ambient mantle, leading many investigators to suggest that arclogites eventually founder back into the mantle[19,32–34]. Foundering would remove the rutiles with high Nb/Ta ratios from the crust, driving what remains of the crust toward the low Nb/Ta signature of the average continental crust. We can estimate the mass of foundered rutile-bearing arclogites using Nb and Ta mass balance. Using the maximum Nb concentration (36 ppm) in these cumulates as a minimum bound of the mass of foundered arclogites, the complementary mass fraction of missing rutile-bearing arclogites would be ~0.3 that of the total continental crust today. If we use the average composition of the rutile-bearing arclogites (Nb = 8.8 ppm, Ta = 0.48 ppm), the foundered mass would be ~2.5 times that of the remaining continental crust, consistent with the estimates based on major element mass balance[35]. Given that our mass balance based on Nb and Ta does not include the more primitive arclogites, our estimates of the mass of the missing arclogite reservoir are minimum bounds.

What is the fate of these foundered arclogites? Garnet pyroxenites have lower solidi than that of peridotite[36,37], and because of this higher fertility, they are more easily melted during mantle upwelling[38–40]. Based on pMELTS simulations[41], we find that the low Mg# arclogites may undergo >70 % melting during decompression from 5 to 2 GPa along a normal mantle adiabat with 1400 °C potential temperature ($T_P$). In contrast, a peridotite

content[27–30]. Although early experimental work suggested a weak pressure effect on rutile solubility in silicate melts[28], more recent experiments show that rutile solubility is significantly reduced at high pressure[29,31]. Therefore, elevated pressures may offset the temperature effect on solubility in less differentiated melts and cause early rutile saturation in continental arc igneous differentiation (Fig. 1a). Using the most recent experimentally

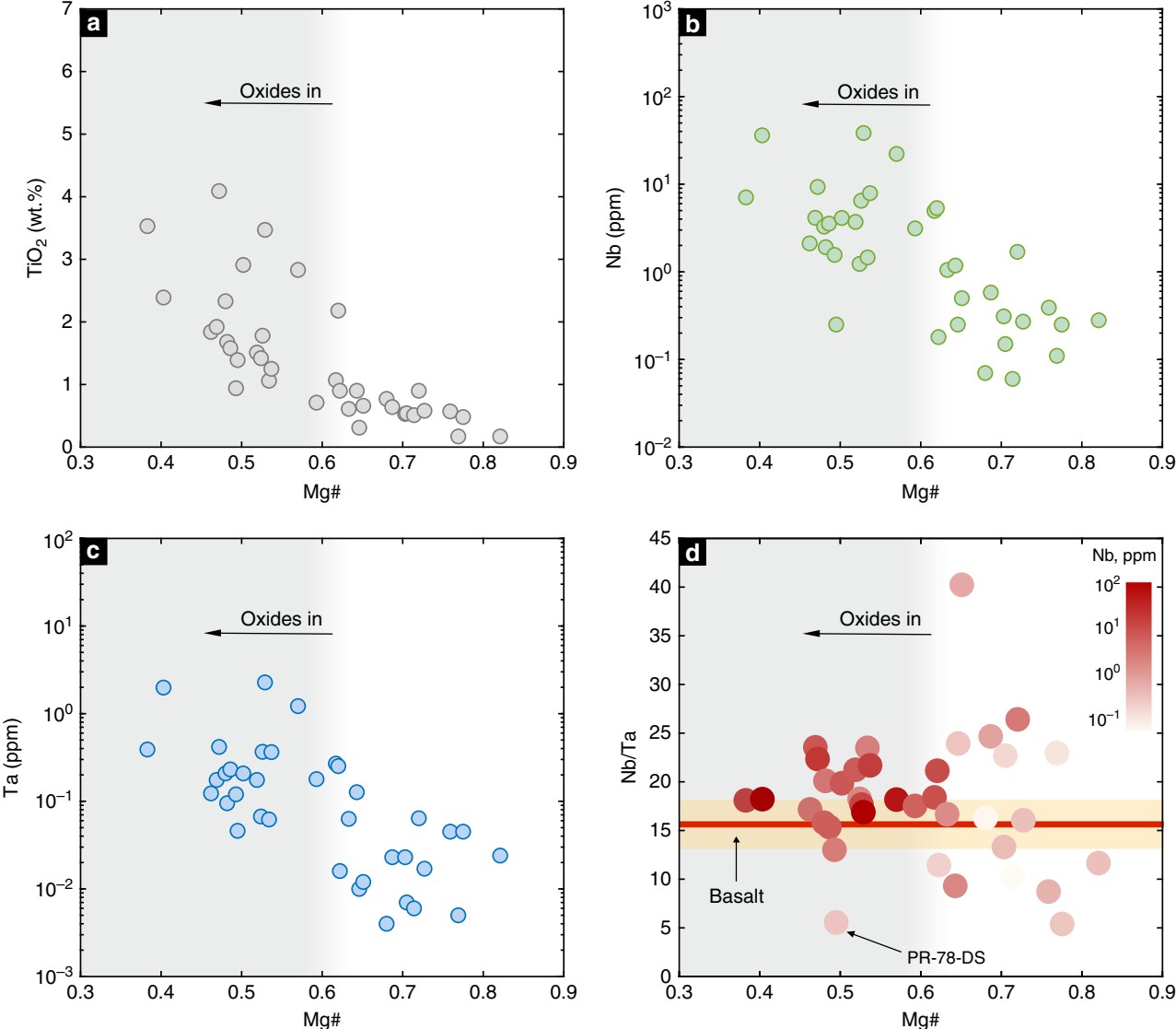

**Fig. 2** Arclogite TiO$_2$, Nb, and Ta concentrations, and Nb/Ta ratio vs. Mg#. The dots in **d** are color-coded by whole rock Nb concentrations. The gray shaded areas indicate the presence of rutile and Fe–Ti oxides in the samples. The red line in **d** shows the average Nb/Ta ratio of global arc basalts and the orange band shows the one standard deviation (1$\sigma$) range

undergoing the same decompression path would only have melted by ~2% (Methods). Rutiles in arclogites may be completely exhausted during early melting (Fig. 4a), producing derivative melts with the high Nb/Ta signatures of their arclogite sources. On the other hand, because garnet remains a major residual phase throughout melting of low Mg# arclogites (Fig. 4a), these high Nb/Ta melts should also have high Dy/Yb ratios. Based on global compilations, we observe a positive correlation between Nb/Ta and Dy/Yb in small volume continental intraplate basalts (Fig. 4b). This is consistent with rutile-bearing arclogites being present in the source region, although the high Nb/Ta in some intraplate basalts may also arise from carbonatite metasomatism in the lithospheric mantle[42]. It is noteworthy that such Nb/Ta–Dy/Yb correlation does not exist in oceanic intraplate basalts (Fig. 4b) despite numerous suggestions that such basalts have pyroxenite in their source regions. The lack of high Nb/Ta in ocean island basalts has also been reported by Pfänder and coworkers[43]. One possible explanation for this difference is that garnet pyroxenites in the source regions of continental intraplate magmas derive from arclogites formed during continent

formation[44]. The source regions of oceanic intraplate magmas might contain eclogites of oceanic crust protoliths and normal Nb/Ta systematics, whereas arclogites may be more common in continental lithospheric mantle. Alternatively, because rutiles can be completely exhausted at low melting degrees (Fig. 4a), rutiles may not survive the prolonged recycling process in the mantle.

Finally, our study also sheds light on the peculiar observation that bulk silicate Earth's Nb/Ta ratio is lower than that of chondritic meteorites[45], which are thought to be the building blocks of Earth. The fact that arclogite Nb/Ta ratios are not significantly higher than that of chondrite[45,46] suggests that the subchondritic nature of the bulk silicate Earth is not due to arclogite formation, but rather a feature that was inherited within the first 100 million years of Earth's history, perhaps via preferential sequestration of Nb into the core during planetary accretion[45,47].

In summary, the formation of rutile-bearing arclogites followed by their foundering into the mantle may be critical to making felsic continental crust. These arclogites have high Fe contents (average total FeO ~ 13 wt.%) and mirror the Fe-

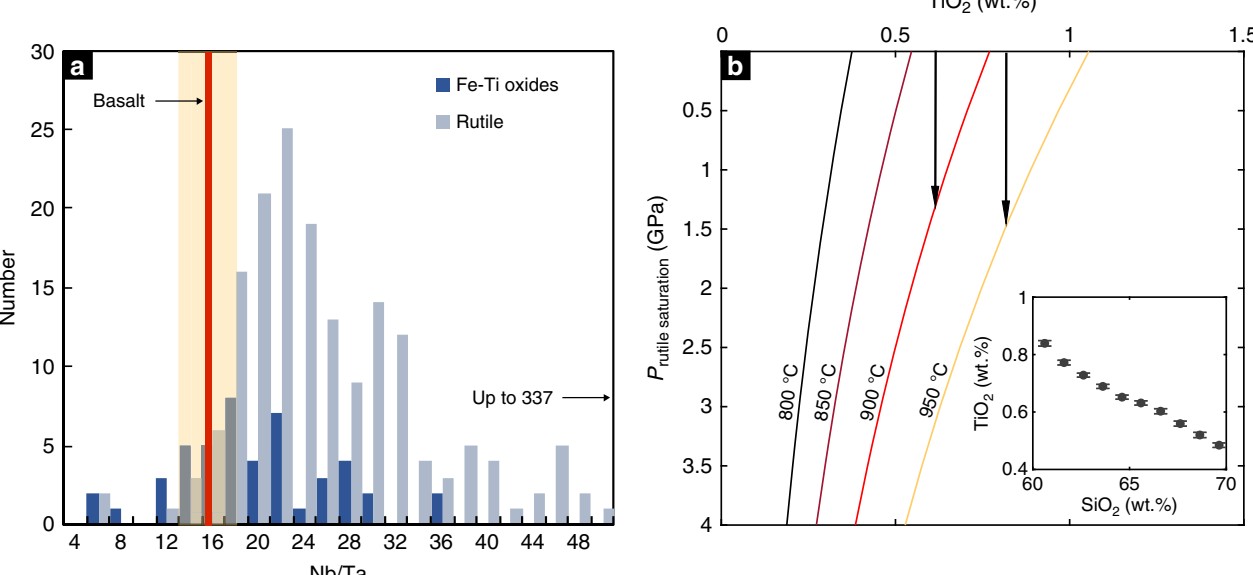

**Fig. 3** Nb/Ta ratio distributions in arclogite rutiles and Fe–Ti oxides and pressure–temperature effects on rutile solubility in differentiated silicate melts. **a** Similar to Fig. 2, the red line shows the average Nb/Ta ratio of global arc basalts (orange band shows the one standard deviation ($1\sigma$)). **b** Calculated rutile saturation pressure as a function of magma $TiO_2$ content ($x$-axis) and temperature (contours). In the rutile solubility calculation, we assumed a water content of 6 wt.%. We used a minimum magma composition parameter (FM, Ryerson and Watson[28]) value of 3 for silicate melts with $60-65$ wt.% $SiO_2$. A higher FM value would require an even greater pressure to saturate rutile (Methods). Inset shows the average $TiO_2$ content as a function of $SiO_2$ content in arc magmas. Error bars are 2 standard errors

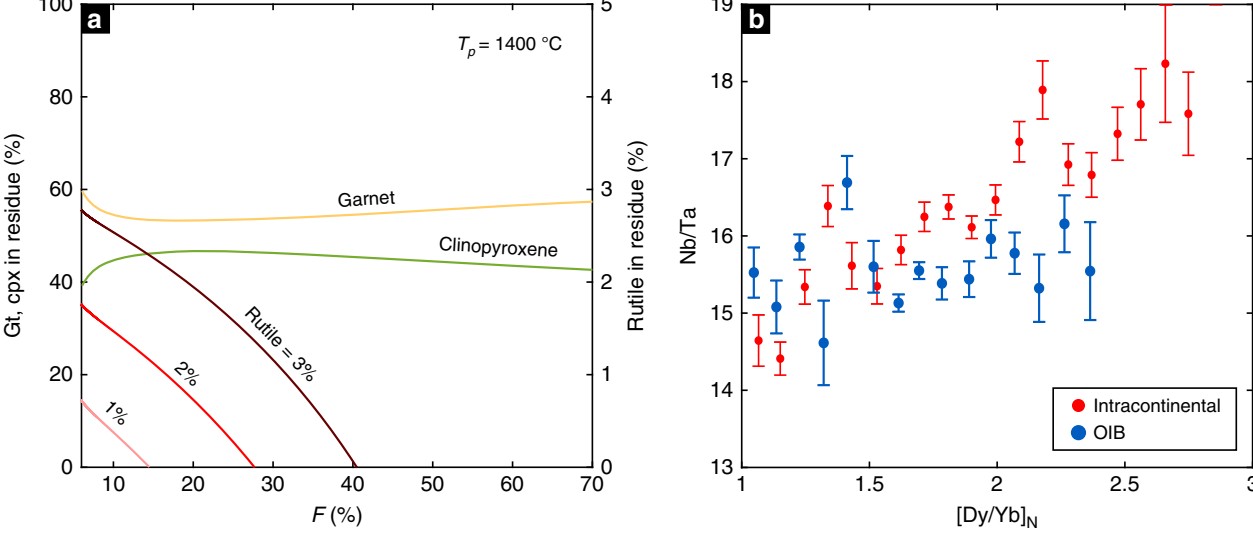

**Fig. 4** Remelting rutile-bearing arclogites in the mantle. **a** Percentages of garnet, clinopyroxene, and rutile in the residue during decompression melting of low Mg# arclogites ($5-2$ GPa, $T_p = 1400\ ^\circ$C). The results are from pMELTS simulations[41]. We considered 1–3% rutile in the solid before melting. The upper bound on the average amount of rutile in the low Mg# arclogites can be calculated from the difference in $TiO_2$ content between the average rutile-bearing and rutile-free arclogites, which is 1.37 wt.%. **b** Nb/Ta vs. chondrite-normalized (Sun and McDonough[64]) Dy/Yb in intracontinental basalts ($n = 4903$) and ocean island basalts ($n = 1915$). Errors are two standard errors (2 se)

depleted, calc-alkaline nature of the continental crust[15,48]. Due to the pressure dependence of rutile saturation in silicate melts (Fig. 3b), the low Nb/Ta ratio of continental crust requires that much of the continents were generated by magmatism in orogenic belts (Fig. 5); today, orogenic magmatism is largely represented by continental arcs[7,19,49]. Because magmatic orogens necessarily develop in regions of convergence, where extensive tectonic compression supports thickened crust, the formation of continental crust may be primarily an outcome of mobile lid

tectonics[50,51] — an Earth's unique surface expression of planetary mantle convection.

## Methods

**Samples.** Our continental arc cumulates are garnet pyroxenite xenoliths collected in the Basin and Range-Colorado Plateau Transition Zone (BR-CP-TZ) of central Arizona, USA[6,15]. These xenoliths were carried to the surface by the volcanic eruptions of Sullivan Buttes latite in Chino Valley and the contemporary Camp Creek latite at ~25 Ma. These deep cumulates have been documented in detail by Erdman and coworkers[6,15] and IEDA online database (https://doi.org/10.1594/

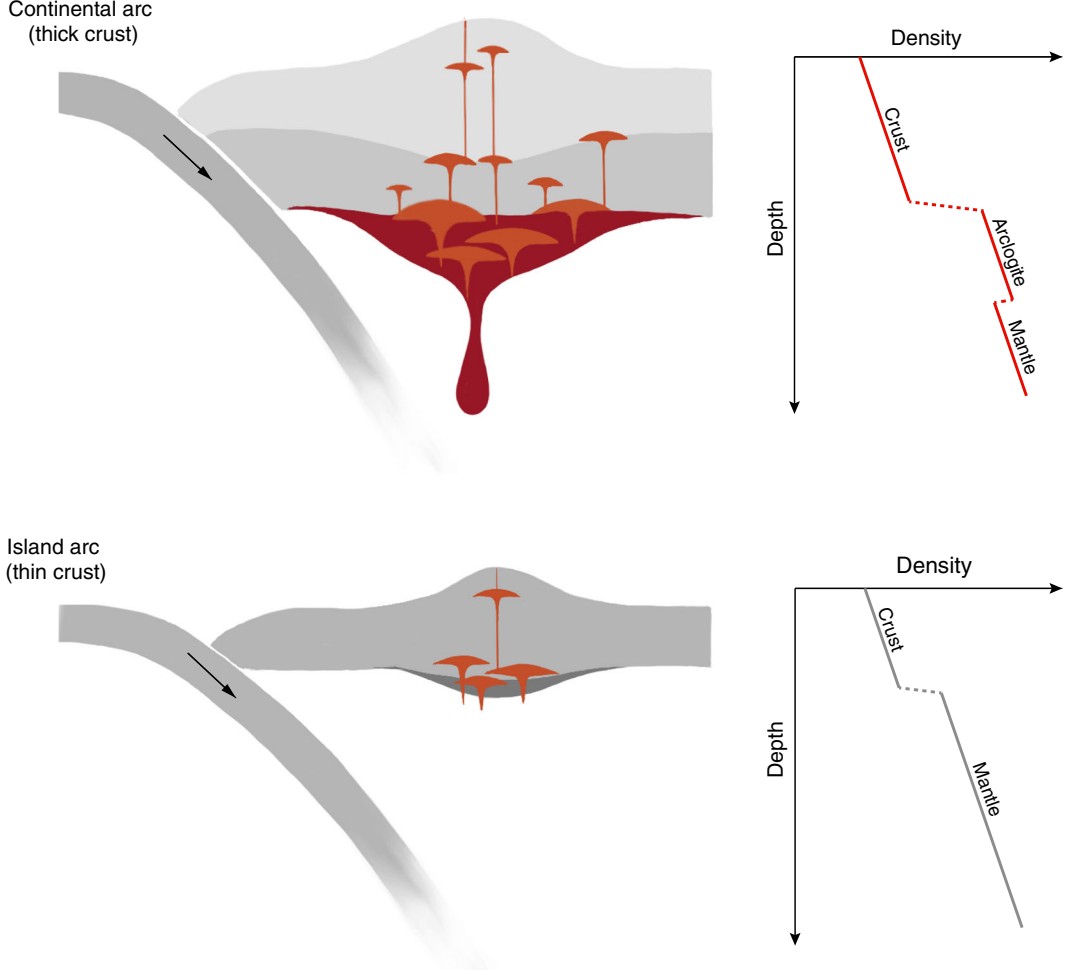

**Fig. 5** Cartoon showing magma differentiation in continental arcs vs. island arcs and schematic crustal density profiles. The dense arclogites beneath continental arcs contain garnet and rutile, hence the dark red color. Co-precipitation of garnet, rutile, and sulfides at the roots of continental arcs removes Fe[13,15,63], Nb (relative to Ta), and chalcophile elements[64] from the magmas and drives calc-alkaline differentiation. In the thickened continental arc crust, magmas may undergo multiple stages of differentiation and produce cumulates at various depths

IEDA/111138). These cumulates have major and trace element compositions identical to arc cumulates in the Sierra Nevada and have ages consistent with Cordilleran arc magmatism in California and Nevada. They represent in situ formed cumulates or tectonically transposed fragments of Sierran arc lithosphere[34].

Garnet and clinopyroxene are the major mineral phases in these cumulates. The garnet-to-clinopyroxene mode ratio generally increases with decreasing whole rock Mg#. Accessory phases including rutile, Fe–Ti oxides, and apatite are only present in the evolved cumulates with Mg# < ~0.6, suggesting their saturation from differentiated magmas. Amphibole is also present in a few samples.

Global arc, ocean island, and small volume intracontinental igneous rock data were extracted from GeoRoc (http://georoc.mpch-mainz.gwdg.de) and are provided in Supplementary Information. MORB data are from Gale et al.[52]. Locations of these samples are plotted in Supplementary Figure 1. To avoid atypical samples, we filtered out 10% samples with the highest Nb/Ta ratios and 10% with the lowest Nb/Ta ratios within each SiO$_2$ and Dy/Yb bins (Fig. 2). In addition, we filtered out all samples with Nb/Ta < 5 as their Nb–Ta systematics may be significantly altered by fluid exsolution processes[53]. Samples with unusually low Nb/Ta may also be contaminated during tungsten carbide grinding processes which may introduce Ta.

Arc basalts, defined as samples with SiO$_2$ = 45–54 wt.% and MgO = 6–15 wt.%, have similar Nb and Ta concentrations as MORB (Supplementary Figure 2). We find no systematic difference in Nb/Ta ratios between arc basalts and MORBs, although arc basalts show more variability (Supplementary Figure 2). Excluding the highest and lowest 10%, we find that MORB and arc basalts have average Nb/Ta ratios of 15.4 ± 1.0 (1$\sigma$) and 15.8 ± 2.4 (1$\sigma$), respectively. These similarities hint that slab contributions may have limited influence on Nb–Ta systematics in most arc lavas, suggesting that intracrustal differentiation processes might be important. The averages Nb/Ta ratios here were calculated using individual samples. Alternatively, one can first calculate the average Nb concentration and average Ta concentration,

and then calculate the average Nb/Ta using the average concentrations. In this way, the average Nb/Ta ratios are 15.8 and 16.0 for MORB and arc basalts, respectively.

**Analytical methods and data reduction**. Whole-rock major element compositions were measured via X-ray fluorescence spectroscopy (XRF) at Washington State University at Pullman. Fresh samples of 10–100 g were crushed and powdered in a ceramic SPEX mill placed in a shatterbox for 5–10 min per sample. Whole rock major element composition data were published at IEDA online database (https://doi.org/10.1594/IEDA/111138).

Whole-rock Nb–Ta analyses were conducted using an Agilent 7900 ICP-MS at the State Key Laboratory of Geological Processes and Mineral Resources, China University of Geosciences, Wuhan. The detailed sample-digestion procedure was modified from ref. [54]. About 50 mg of sample powder was weighed into a Teflon bomb, and then 1 ml of concentrated HNO$_3$ and 1 ml of concentrated HF were added. The sealed bomb was heated at 190 °C in oven for 72 h. The bomb was opened, and the solution evaporated at ~120 °C to dryness. This was followed by adding 1 ml concentrated HNO$_3$ and evaporating to dryness again. The resultant salt was re-dissolved by adding ~3 ml of 30% HNO$_3$, and resealed and heated in the bomb at 190 °C for ~24 h. The final solution was diluted to ~100 g with 2% HNO$_3$ for ICP-MS analysis. Six BHVO-2 replicate analyses imply that the precision (2 RSD%) is better than 5%. Our measured Nb/Ta ratios for six reference materials are compared with GeoRem recommended values in Supplementary Figure 3. The relative difference between our measured values and GeoRem recommended values is less than 3%. The data are provided in Supplementary Data 1.

Cumulate whole rock trace elements were also analyzed by laser-ablation inductively coupled plasma mass spectrometry (LA-ICP-MS) on lithium metaborate-fused glass discs for XRF analyses at Rice University. The data were published at https://doi.org/10.1594/IEDA/111138. The solution Nb ICP-MS data

agree well with the LA-ICP-MS Nb data (Supplementary Figure 4); the solution ICP-MS Ta data agree with the LA-ICP-MS Ta data for samples with > 0.05 ppm Ta (Supplementary Figure 4). There is significant discrepancy between the two methods in samples with <0.05 ppm Ta, in which case the Ta concentration is close to the detection limit of the LA-ICP-MS measurements of fused glass diluted by lithium metaborate.

Nb, Ta, Zr, Hf, Mg, Fe, and Ti concentrations in rutile and Fe–Ti oxides were analyzed using a Thermo Finnigan Element2, a single-collector, sector field, ICP-MS, coupled to a New Wave Research, frequency-quintupled (213 nm) Nd:YAG laser system in the Earth, Environmental and Planetary Sciences Department at Rice University. We used a spot size of 30 μm diameter throughout the analysis, and measured $^{25}$Mg, $^{57}$Fe, $^{49}$Ti, $^{90}$Nb, $^{93}$Zr, $^{178}$Hf, and $^{180}$Ta. Helium (775 ml/min) was used as a carrier gas to transport the aerosol from the sample cell to the ICP. The laser was operated with fluence of 15 J/cm$^2$ and a repetition rate of 10 Hz. The measurements were carried out in low mass resolution mode ($m/\Delta m \sim 300$). Sensitivity was ~30,000 cps/ppm for La on a 55-μm spot. An ~10 s background signal was analyzed before ablation. This background signal was subtracted from the ablation signal. Torch positions and sample gas flow rate were tuned to minimize oxide production ($^{238}$U$^{16}$O/$^{238}$U < 0.5%) before each analytical session. We used USGS glass standard BHVO-2G as the external standard and measured it twice every hour for calibration. Following Liu et al.[55], we used the bulk major element oxides, $TiO_2 + FeO + MgO$ in the case of oxides, as the internal standard (assumed to be 100 wt.%) to calculate the concentrations of Nb, Ta, Zr, and Hf. The data are provided in Supplementary Data 2. We note that the difference in matrix between our glass standards and the metal oxides may lead to errors in the measured oxide Nb and Ta concentrations. But this matrix-dependent Nb/Ta fractionation appears to be limited[26,56].

Titanite is seen in some of our low Mg# arclogite samples. Titanite is also a Nb-, Ta-compatible phase. However, titanite appears to prefer Ta over Nb even at low temperatures[21,57]. Our measurements of Nb/Ta in magmatic titanites from the Peninsular Ranges Batholith[58] (Supplementary Data 2) confirm their low Nb/Ta (average Nb/Ta = 7.4 ± 3.8, 1$\sigma$). Thereby, titanite fractionation should increase Nb/Ta in the melts and decrease Nb/Ta in the cumulates.

**Calculation**. The mass of recycled rutile-bearing arclogites can be calculated by adding arclogites back to the remaining continental crust until the bulk Nb/Ta equals average basalt value (15.8). We used the continental crust Nb and Ta compositions from Rudnick and Gao[1]. For the recycled rutile-bearing arclogite composition, we considered the average low Mg# arclogites and the arclogite with the highest Nb and Ta concentrations as two possibilities. Nb and Ta mass balance is given by Eq. 1:

$$\frac{C^{Nb}_{cumu} \times m_{cumu} + C^{Nb}_{cc} \times m_{cc}}{C^{Ta}_{cumu} \times m_{cumu} + C^{Ta}_{cc} \times m_{cc}} = 15.8, \quad (1)$$

where $C^{Nb}_{cumu}$ and $C^{Ta}_{cumu}$ are Nb and Ta concentrations in the recycled rutile-bearing arclogites (cumulates), $C^{Nb}_{cc}$ and $C^{Ta}_{cc}$ are the Nb and Ta concentrations of the remaining continental crust, and $m_{cumu}$ and $m_{cc}$ are the masses of the recycled rutile-bearing arclogites and the remaining continental crust, respectively.

We calculated rutile solubility in silicate melts based on the most recent experimental calibrations[29], which accounts for the effects of temperature, pressure, melt composition (FM), and water content:

$$\ln(TiO_2)_{melt} = \ln(TiO_2)_{rutile} + 1.701 - \left(\frac{9041}{T}\right) \\ - 0.173P + 0.348FM + 0.016H_2O, \quad (2)$$

where $TiO_2$ and $H_2O$ are in wt.%, $T$ and $P$ are in K and GPa, respectively. We assumed the water content in differentiated arc magmas is 6 wt.%. FM is a melt composition term calculated as:

$$FM = \frac{1}{Si} \times \frac{Na + K + 2 \times (Ca + Fe + Mn + Mg)}{Al}, \quad (3)$$

where Si, Na, K, Ca, Fe, Mn, Mg, and Al are cation fractions in the melt. This melt composition term FM was also used by Ryerson and Watson[28] and Hayden and Watson[27]. The FM value can be evaluated from magma compositions as a function of $SiO_2$ (Supplementary Figure 5). At $SiO_2 = 60-65$ wt.%, the average value of FM is between 3 and 4. In the paper, we used the lower FM value of 3 to calculate rutile solubility in a silicate melt. Using the higher FM value of 4 will result in a higher pressure (>3 GPa) to saturate rutile in a silicate melt with 60−65 wt.% $SiO_2$ at 900 −950 °C (Supplementary Figure 6).

We modeled peridotite and pyroxenite decompression melting from 5 to 2 GPa along the mantle adiabat (potential temperature $T_p = 1400$ °C) by pMELTS[41]. The relationships between melt fraction and pressure are shown in Supplementary Figure 7. We used the average rutile-bearing garnet pyroxenite (Mg# < 0.6) composition as the starting composition for pyroxenite, and average depleted mantle composition[59] as the starting composition for peridotite.

## Data availability

The data reported in this study are provided in Supplementary Dataset 1 and Supplementary Dataset 2. The compiled arc, intracontinental and ocean island magma data are provided in Supplementary Dataset 3.

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

## Acknowledgements

This work was supported by U.S. National Science Foundation grants to Lee (OCE-1338842, EAR-1347085). K.C. is grateful to the National Nature Science Foundation of China (41703034). We thank Dr. Xun Yu for discussion and the constructive review comments from Mihai Ducea, Massimo Chiaradia, and Jörg A. Pfänder.

## Author contributions

M.T. designed the project, conducted in situ measurements, compiled the data and carried out the simulations. K.C. conducted the solution trace element measurements. M.T. and C.-T.A.L. wrote the manuscript. All authors participated in data interpretation.

## Additional information

**Competing interests:** The authors declare no competing interests.

