## [Peer Review File · Nature Communications]

Reviewers' comments:

Reviewer #1 (Remarks to the Author):

I fundamentally agree with this manuscript and others that see a clear correlation between making silica -rich magmatic products by differentiation processes in deeper crustal environments of thick crust. Among the lesser exploited controversies that could be built into this hypothesis, is that the Nb/Ta paradox of Rudnick and others would be solved by the presence of rutile and other iron oxides in dense cumulates. So I would totally give the green light to the publication of this manuscript because it is one more link to the solving of the continental origin question. A few comments below are intended to help clarify some of the argument – with or without them, the paper is strong enough to be published after some minor adjustments. Overall I'd give this paper a solid 9.5 out of 10 for importance and delivery. I wish I wrote it.

I will start with a weird philosophical issue. One aspect that I myself struggled with over time is how to make continents only in.... continental arcs; at some point this had to start without continental arcs being present. Stating that you need thick continents to make continents requires an additional explanation. I suppose the actual requirement is the presence of deep seated fractionation as opposed to what takes place at modern island arcs where most fractionation (the filter imposed by the upper plate) takes place at shallow levels (a few kbars). Either early on continents formed by some other deep seated form of fractionation (see comment below) which later gave way to the processes envisioned in this paper and elsewhere, or they formed gradually from progressively thicker continental arcs being available in various places.

I would argue that despite the statement at the end of the abstract , this process is not necessarily plate tectonic related – early such cumulates mimicking continental arcs could have been thick oceanic plateaus or some other form of cumulates that satisfy primarily the silica enrichments differentiation trends @ high pressures (they happened at depths of 40 km or more). Garnet as well as other phases need great depths and they are also forcing the fractionated magma to become more silicic. In a sense, Fe-Ti oxides from the magnetite ilmenite family are not really limited to high pressure arc type fractionation – they do in fact form at shallower levels as well. But rutile may be the big one capturing the Nb/Ta signal. And my recollection (and I did not find this observation in this ms) is that the Sierran low Mg pyroxenite (arclogite) xenoliths and equivalents in South America are characterized by garnet FULL of rutile inclusions. It is a signature of these rocks in thin sections – the needles of rutile included in the garnet.

Line 95: This is a strange sentence in that the references cited have nothing to do with "these pyroxenites" - they come from a totally different place. More importantly, I am not sure anybody has proven that the central Arizona xenoliths are arc cumulates -. And if they are, which arc? The Laramide arc? Some early Cenozoic arc which has very few upper crustal plutons exposed in the geology of Arizona? One more minor thing- you are pretty far from the Colorado plateau. You could have just as easily called them "from the edge of the Sierra Nevada given all the extension that took place since. It gives the impression that they are somehow related to the Colorado plateau which I believe they are not. This needs cleaned up a bit or some additional geologic explanation dumped in the supplementary files. These rocks sure look like the Sierran cumulates but their belonging to a specific magmatic arc is still unresolved. Perhaps Alan Chapman's new data suggesting they are displaced from California during the shallow bulldozing event? Whatever the explanation, it will not change the relevance of the paper, but some people will ask.

Line 165 and beyond. The referneces 28, 41-42 could not have been more annoying. That is because this has been proposed previously – see my papers – Ducea, 2002, Ducea and Saleeby, 1998, Saleeby et al., 2003 (not cited) while the classic and oft cited papers by Kay and Kay and ESPECIALLY Rudnick are not really about garnet pyroxenites at all. I challenge you to find any reference to garnet pyroxenites (which is something that our group and Cin Ty's have pioneered based on Sierran xenoliths) in Rudnik's paper which is a vague argument paper about missing reservoirs. Incidentally, I found the citations in this paper to be sloppy overall. Perhaps the first author does not know all these things, but the senior author should intervene here.

Why just bring the word arclogites late in the paper? It is a cute word and if used at all (and as far as I am concerned, go for it), bring it up earlier and use it.

Line 206 – these arclogites may be important in the deep lithosphere, it is stated here. Or not. The whole idea is that they founder in the mantle, isn't it? So they are no longer part of the lithosphere? Hence the missing reservoir.

Overall I really enjoyed the manuscript

Reviewer #2 (Remarks to the Author):

The ms by Tang et al. "Building continents in magmatic orogens: Nb/Ta systematics in arc magma differentiation and the role of arclogites" presents and discusses data on Nb/Ta systematics in arc magmas and the implications for the mechanisms of continental crust formation.

I find the ms well written and the topic addressed is suitable for a broad audience like that of Nature Communications. The authors convincingly show that Nb/Ta systematics of arc magmas differ between those from thin and thick arc settings. Correlations between Nb/Ta and Dy/Yb are taken to indicate that the low Nb/Ta values of the thick continental crust are produced in concomitance with deep level fractionation of arc magmas (in the stability field of garnet) in the thick arc setting compared to the thin one. Through analyses of rutile (LA-ICPMS) and P-dependent solubility models of rutile they also show that the potential responsible for the low Nb/Ta of the continental crust is rutile in the garnet-pyroxene cumulates that are the product of the deep crustal fractionation.

Overall I find this ms very interesting for the understanding of continental crust formation mechanisms and that it can be accepted after very minor points are addressed, as outlined below and in the annotations of the attached file:

1. I would like to invite the authors to consider also the role of zircon and titanite in controlling the Dy/Yb ratio and therefore its correlation with Nb/Ta. This possibility may be especially important concerning high SiO₂ rocks for which fractionation of these two minerals is highly probable. The main line of reasoning would not change essentially because Dy/Yb would anyway be controlled by crystal fractionation.
2. You should specify on which rutiles you performed LA-ICPMS analyses: are these the rutiles from the cumulates or others? Is there any meaning for the large Nb/Ta range of values of these rutiles? i.e., dependence on setting, rock type, etc. in the case these rutiles are from different rocks and/or settings or also if they are all from cumulates? For this reason it is important to specify the petrogenetic context of the rutiles that you analysed.
3. Lines 136-138: "That rutile and Fe-Ti oxides are saturated as magmatic phases only in the low Mg# cumulates, suggests their crystallization from evolved and hence cooler magmas as the low Mg# cumulates are thought to be complementary to andesitic to dacitic magmas": Perhaps it would be good to have some firmer evidence for this (i.e. modeling based on Mg# of cumulates versus andesite-dacite) or at least a reference if there is one.

A few minor edits are reported in the attached file

I think that this ms can be an influential work for the understanding of the mechanisms of continental crust formation and will be of interest for a broad community of the Earth Sciences.

Reviewer #3 (Remarks to the Author):

General remarks

The manuscript aims to provide a model to explain the overall low Nb/Ta in the continental crust by intracrustal Nb/Ta fractionation, where Ti-rich phases (rutile) play a major role. It is of great importance for the geochemistry community that deals with the differentiation of the solid Earth in a wider sense, and the outcome is original and novel and supported by the data. The manuscript is largely well and convincingly written and clearly organized, a few exceptions are marked in the PDF (towards the end, the text gets a bit more difficult to read and to follow, and a few slips of the pen need to be removed). The scientific idea is smart, the approach and concept are appropriate, and the results are significant for a better understanding of continental crust formation starting from an originally ultramafic source rock. With this regard, it might influence thinking in this field of research. The observation that on a very global scale MOR basalts and island arc basalts share about identical Nb/Ta, and the derived conclusion that Nb/Ta differentiation to make low Nb/Ta continental crust therefore needs to be an intracrustal process is highly important. Based on this conclusion the manuscript, to my knowledge for the first time from this perspective, provides a very plausible mechanism to explain the overall accepted but rarely questioned low Nb/Ta in continental crust. With this respect it is a very important contribution to the "Nb paradox" discussion. The methodological approach is sound, the own data support the conclusions, and the data taken from the Georoc database have been filtered, and potential corruptions of Nb/Ta ratios were briefly discussed in the methods chapter. The scientific outcome is qualitatively supported by the data, but the manuscript misses a desirable quantitative support of the major conclusion (see comment to line 63-65 below).

Several weak points need to be also addressed, although they do not interfere the overall importance of the manuscript. So, the actual knowledge regarding the "Nb paradox" discussion is not described in sufficient detail, or presented in a too oversimplified way (see below), possibly to more strongly justify the need of the present work. This must be improved but will not detract the overall importance of the study. Then, the very global aspect addressed, i.e. transferring the process to the formation of continental crust in general, seems not to be fully justified. Clearly island arc and passive continental margin magmatism is important in building the continental crust, but given that the key process does obviously only occur in very thick active continental margins, the question arises to what extent this process can be regarded as the fundamental one that solely controls the Nb/Ta distribution in the continental crust. Possibly the authors should reduce their smart and well supported model to "some" active continental margin magmas with a (very) specific setting (i.e. very thick arc crust, what happens before and how is this thickness achieved?). Except, they can provide additional arguments that show that continental crust formation does only occur in very thick active continental margins. Beside the overall plausible explanation for low Nb/Ta in the continental crust, the authors describe in brief the fate of arclogites by suggesting foundering them back into the mantle. Although this makes sense, the expected high Nb/Ta in the order of 19 or more (see below)

are not observed in any ocean basin derived basalt. This is evident from the data presented in the manuscript (Fig. 4b), as well as from the literature. This is conflicting and needs to be resolved.

The cited literature is appropriate, but some rather fundamental and important papers are missing (e.g. Münker et al. 2003, *Science*, 301, 84-87; Weyer et al. 2003, *EPSL*, 205, 309-324; Jochum et al., 2000, *Meteoritics & Planetary Sciences*, 35, 229-235). These amongst others will help to complement the description of the "Nb-paradox".

In summary, the presented model is a plausible and novel solution for broadly accepted low Nb/Ta in the continental crust, as it ascribes the Nb/Ta differentiation to intracrustal processes. I.e. it moves the location of global Nb/Ta fractionation from mantle settings to crustal settings, which is a novel perspective that makes sense. The underlying process is plausibly derived from own and literature data. What remains unresolved or at least vague is the fate of the arclogites. Foundering them back into the mantle seems unlikely as ocean basalts are too low in Nb/Ta. However, the authors themselves provide a potential solution to this problem by linking the arclogites to continental intraplate magmatism, as continental basalts share the required high Nb/Ta. Strengthening this issue on the expense of the foundering hypothesis will result in an overall plausible model and a consistent manuscript.

I strongly recommend publication of the manuscript after having refined the above mentioned issues and after having carefully addressed the points listed below and in the PDF file.

Specific comments

Lines 21-23. This statement is not valid as it is too oversimplified, but it is an important prerequisite to justify the major conclusions of this manuscript. The "state-of-the art" regarding the "Nb paradox" should be outlined more comprehensively. Depleted mantle can be as low as 10 in Nb/Ta (Weyer et al. 2003, *EPSL*, 205, 309-324), MORB is around 14 but with distinct variations that result from partial melting of different sources (see also Weyer et al. 2003), OIBs are 15-16 (Pfänder et al., 2007, *EPSL*, 254, 158-172) and intraplate basalts are higher, between 15.0 - 19.1 (Pfänder et al., 2012, *CMP*, 2012, 232-251). The variation in Nb/Ta in silicate rocks is undoubtedly low, but resolvable, and this is due to partial melting, but primarily a result of different sources involved during melt formation. This needs to be figured out more clearly.

Lines 35-36. Due to mass balance considerations, this seems to be unlikely on a global scale. How can arclogites compensate the volume of the whole continental crust? Where should these be? If such a statement is presented, it should be quantified by model calculations to give a realistic idea about the proportions needed to compensate the Nb deficit of the continental crust.

Line 38-39. Clearly continent formation is linked to plate tectonics, this is trivial.

Line 44-45. This needs references. Stolz et al. 1996 (*Geology*, 24, 587-590) reported elevated Nb/Ta in K-rich arc lavas (25 – 33).

Fig. 1a: The backbone of the manuscript are the low Nb/Ta in a little more than four “bins”, i.e. comparatively little data points averaged from a number of samples from Georoc. It needs to be resolved, which locations and settings these samples represent, are they representative to place a global scale conclusion, or are these locations “exotic” ones? In fact, only very few data points have Nb/Ta as low as continental crust, and if so, at higher or even much higher SiO₂. This rises the question whether this model is appropriate to derive a model for low Nb/Ta in continental crust in general. Rutile fractionation in thick arc crust seems to be undoubtedly an important process, the database is sound, the trends in Fig. 1 are clear and the overall conclusion is correct, but is it valid on a global scale or rather in some exceptional cases/settings where highly differentiated melts evolve in very thick active continental margin crust? What is the role of amphibole fractionation in forming continental crust with low Nb/Ta as suggested by Li et al. (2017) (*Journal Petrology*, as cited in the manuscript)?

Line 63-68, Caption Fig. 1. The volumes of highly differentiated arc magmas and arc basalts will be very different in continental arcs, with arc basalts dominating. I'd claim that to produce crust with Nb/Ta ~11 and SiO₂ ~62wt%, unrealistically high volumes/masses of differentiated magmas are required. Please provide calculated mixing lines and denote the portion of differentiated melt added on the tick marks. Also mark the end-member compositions with symbols in Fig. 1a.

Line 116. ... until after ...

Line 117-118. This is not only valid for the low Mg# cumulate samples, but also for high Mg#, although their average might be lower? Is there really a difference?

Line 121-123. Provide an average Nb/Ta for the high Mg# samples, and 1s. What is the concentration weighed total cumulate average Nb/Ta value? Please denote.

Line 124ff. It is known since long that if present, Ti-bearing phases, in particular rutile, control the HFSE budget of a system. Needs no in depth evaluation, possibly cite one or two papers that provide consistent partition coefficients for Nb and Ta between rutile, Fe-Ti oxides and mafic magmas. Rudnick et al. (2000) (*Science*, 287, 278-281), and later Aulbach et al. (2008) (*Nature Geosciences*, 1, 468-472) both identified rutile as dominant host for Nb and Ta in subducted eclogites and in the subcontinental lithospheric mantle.

Line 132-134. According to the equation given in the cited paper (Xiong et al., 2011), DNb/Ta becomes >1 only in exceptional cases, i.e. for low water contents (<4-5wt%), at relatively low temperatures (<1000°C) and high pressures (>2 GPa). This results in very low geothermal gradients (<15°C/km), to my opinion unrealistic in active continental margins with a long lasting magmatic history. Therefore, the provided explanation for the discrepancy between experimental data and observation is unsatisfying. The authors should test and more comprehensively discuss what the causes of this contradiction could be. It seems also unrealistic to assume such low water contents in highly differentiated arc magmas.

Lines 168-171. Unclear. Please explain properly and comprehensively what you did and what the assumptions are.

Line 171-176. A foundering model is favored to remove the garnet-pyroxenite cumulates with high Nb/Ta from the base of the thickened crust. The mass estimate based on averaged concentrations seems to yield a mass of foundered material of at least 2.5 times the mass of the continental crust. With this respect, this model resembles the eclogite recycling model of Rudnick et al. (2000) (see above). Given that the recycled material of either subducted or foundered garnet-pyroxenite ("eclogite") has a lower solidus temperature than ambient mantle peridotite (as outlined in lines 188-190), partial melts of such sources should be at least partly sampled by oceanic magmatism such as OIBs. Those should therefore indicate elevated Nb/Ta, i.e. >18.8 if the averaged cumulate Nb/Ta of this study is taken as source value, as partial melting, even in the presence of rutile, will further enlarge Nb/Ta in the melt relative to the source. This, however, is not the case as is visible from Figure 4b (for high precision Nb/Ta in OIBs and melting modelling see also Pfänder et al., 2007, EPSL, 254, 158-172). Any other option of storage? Is the foundering necessarily required?

Line 199-200. I would engage the authors to undertake melting calculations to verify the qualitative statements made. As outlined before, Nb/Ta in melts from foundered garnet-pyroxenite with $\text{Nb/Ta} \sim 18-19$ should be at least equal or higher, which is not observed taking the sample compilation in Fig. 4b and literature data (Pfänder et al., 2007).

Line 200. The cited reference (Pfänder et al., 2012) invokes carbonatitic melts to explain elevated Nb/Ta in continental basalts, not pyroxenite melts, which, however, might be a good alternative based on the observations made here and as discussed in lines 202-206.

Line 467-471. Extended Data Figure 3b displays that at overall low Ta contents the solution based measurements provide significantly lower concentrations than the LA-ICP-MS measurements. The explanation for this is weak. Explaining it by low count rates for LA-ICP-MS is unsatisfying. This should result in the opposite. Either this is background or an isobaric interference during LA-ICP-MS, or the solution method loses Ta, possibly by adsorption to teflon vial walls, why Munker et al. (2001) (G-cubed, Vol. 2, 10.1029/2001GC000183) have suggested to use traces HF in all acid steps during HFSE analyses in solution based methods. It is crucial to know which of the methods is appropriate.

Lines 488-489. Without being an expert in LA-ICP-MS, I'd expect that a silicate glass standard will release and fractionate different elements in a different manner than oxides, with different bond strengths and crystal lattice geometries. Are there no oxide standards (rutile) available? Calibrated natural rutile or Fe-Ti oxides to minimize matrix effects?

Line 498. Are errors of the coefficients of the equation given in the original publication?

Line 504. This SiO_2 holds for erupted arc magmas (from which the curve in Extended Data Figure 4 is derived), but prior to fractionation and during fractionation, it should be lower. Is there a lower SiO_2 -limit in the fractionating magma where the pressure needs to be unrealistically high to become rutile to be a liquidus phase?

Extended Data 1: Would suggest to add the Nb/Ta ratios of the standards and compare them to the ratios calculated from the recommended values. An additional comparison can be made to Pfänder

et al. (2007; see above), who provides Ta-ID concentration values for BHVO-2 and high-precision Nb/Ta for this standard.

Revision notes

We greatly appreciate the constructive and detailed comments from the reviewers. We agree with the reviewers and revised the manuscript accordingly. Briefly,

- We added a new panel in Fig. 1 to show the systematic difference in Nb/Ta in differentiated magmas between thick continental arcs and thin island arcs;
- We added two new figures (one cartoon in the main text and one figure in the methods showing the accuracy of our solution analyses);
- We fixed all the reference issues mentioned by the reviewers;
- We added some discussions about the implications of our work on the subchondritic Nb/Ta of bulk silicate Earth, as suggested by the third reviewer;
- We added some discussions (with new data) about the potential effects of zircon and titanite on REE and Nb/Ta fractionation in the method, as suggested by the second reviewer;
- We provided more details about our Nb/Ta mass balance calculations in the methods, as suggested by the third reviewer;
- We fixed the typos and rephrased the inaccurate expressions pointed out by the reviewers.

Below is our point-by-point response. Changes in the manuscript are highlighted in red.

Reviewer #1 (Remarks to the Author):

I fundamentally agree with this manuscript and others that see a clear correlation between making silica - rich magmatic products by differentiation processes in deeper crustal environments of thick crust. Among the lesser exploited controversies that could be built into this hypothesis, is that the Nb/Ta paradox of Rudnick and others would be solved by the presence of rutile and other iron oxides in dense cumulates. So I would totally give the green light to the publication of this manuscript because it is one more link to the solving of the continental origin question. A few comments below are intended to help clarify some of the argument – with or without them, the paper is strong enough to be published after some minor adjustments. Overall I'd give this paper a solid 9.5 out of 10 for importance and delivery. I wish I wrote it.

I will start with a weird philosophical issue. One aspect that I myself struggled with over time is how to make continents only in... continental arcs; at some point this had to start without continental arcs being present. Stating that you need thick continents to make continents requires an additional explanation. I suppose the actual requirement is the presence of deep seated fractionation as opposed to what takes place at modern island arcs where most fractionation (the filter imposed by the upper plate) takes place at shallow levels (a few kbars). Either early on continents formed by some other deep seated form of fractionation (see comment below) which later gave way to the processes envisioned in this paper and elsewhere, or they formed gradually from progressively thicker continental arcs being available in various places.

Yes, we agree and the point we want to make is that high pressure differentiation is key to making the continental crust. High pressure intracrustal differentiation requires the upper plate to be thick but not necessarily felsic or of modern continental crust compositions, so ultimately the question is what drives crustal thickening. Today, synmagmatic crustal thickening is mostly seen in magmatic orogens such as continental arcs (e.g., Andes) and continental collision zones (e.g., Tibet). While this model may explain the mechanism of continental crust formation in the Phanerozoic, whether it also applies to continent formation in the Archean is an open question.

I would argue that despite the statement at the end of the abstract, this process is not necessarily plate tectonic related – early such cumulates mimicking continental arcs could have been thick oceanic plateaus or some other form of cumulates that satisfy primarily the silica enrichments differentiation trends

@ high pressures (they happened at depths of 40 km or more). Garnet as well as other phases need great depths and they are also forcing the fractionated magma to become more silicic. In a sense, Fe-Ti oxides from the magnetite ilmenite family are not really limited to high pressure arc type fractionation – they do in fact form at shallower levels as well. But rutile may be the big one capturing the Nb/Ta signal. And my recollection (and I did not find this observation in this ms) is that the Sierran low Mg pyroxenite (arclogite) xenoliths and equivalents in South America are characterized by garnet FULL of rutile inclusions. It is a signature of these rocks in thin sections – the needles of rutile included in the garnet.

We agree that crustal thickening may not necessarily happen in convergent boundaries. Oceanic plateaus are thicker than surrounding oceanic crust, but their average crustal thickness is only 10-30 km (Kerr, 2003), far less than that of magmatic orogenic belts that develop at plate convergent boundaries (60-80 km). This is likely due to the lack of tectonic compression in intraplate settings, which plays a significant role in thickening the crust at convergent boundaries. With only 10-30 thick crust, oceanic plateaus cannot stabilize garnet at the base. In addition, hydrous conditions also help to stabilize garnet in magmatic differentiation (Alonso-Perez et al., 2009; Green, 1972). Intraplate magmas are generally drier than arc magmas, which requires even greater pressure for garnet saturation. Thereby, it appears that convergent boundaries that form in plate tectonics regime are far more efficient in making thick crust and stabilizing garnet during differentiation.

Rutile is the only phase that has high Nb/Ta ratios and high Nb, Ta and Ti contents in these high pressure cumulates. Indeed, many rutile crystals do exist as inclusions in garnet and clinopyroxene. We have added this observation in the revised manuscript (line 118-119).

Line 95 (now in Line 100): This is a strange sentence in that the references cited have nothing to do with "these pyroxenites" - they come from a totally different place. More importantly, I am not sure anybody has proven that the central Arizona xenoliths are arc cumulates -. And if they are, which arc? The Laramide arc? Some early Cenozoic arc which has very few upper crustal plutons exposed in the geology of Arizona? One more minor thing- you are pretty far from the Colorado plateau. You could have just as easily called them "from the edge of the Sierra Nevada given all the extension that took place since. It gives the impression that they are somehow related to the Colorado plateau which I believe they are not. This needs cleaned up a bit or some additional geologic explanation dumped in the supplementary files. These rocks sure look like the Sierran cumulates but their belonging to a specific magmatic arc is still unresolved. Perhaps Alan Chapman's new data suggesting they are displaced from California during the shallow bulldozing event? Whatever the explanation, it will not change the relevance of the paper, but some people will ask.

We have now replaced the wrong references by Erdman et al. (2016) and Tang et al. (2018). We agree that the origin of these cumulates are being debated. We have removed "Colorado plateau" in the text so as not to cause confusion. In the Methods section, we added several lines to explain the possible origins of these cumulates (line 472-475).

Line 165 (now in Line 172) and beyond. The referneces 28, 41-42 could not have been more annoying. That is because this has been proposed previously – see my papers – Ducea, 2002, Ducea and Saleeby, 1998, Saleeby et al., 2003 (not cited) while the classic and oft cited papers by Kay and Kay and ESPECIALLY Rudnick are not really about garnet pyroxenites at all. I challenge you to find any reference to garnet pyroxenites (which is something that our group and Cin Ty's have pioneered based on Sierran xenoliths) in Rudnik's paper which is a vague argument paper about missing reservoirs. Incidentally, I found the citations in this paper to be sloppy overall. Perhaps the first author does not know all these things, but the senior author should intervene here.

We have now fixed the references.

Why just bring the word arclogites late in the paper? It is a cute word and if used at all (and as far as I am concerned, go for it), bring it up earlier and use it.

Thanks for the suggestion, and we have now brought up the word arclogite earlier in the paper.

Line 206 (now in Line 213) – these arclogites may be important in the deep lithosphere, it is stated here. Or not. The whole idea is that they founder in the mantle, isn't it? So they are no longer part of the lithosphere? Hence the missing reservoir.

It is possible that much of the foundered rutile-bearing arclogites may not be recycled into the convective or deep mantle but form part of the lithospheric mantle, or at least most of the rutiles were melted away before the arclogites reached the deep mantle. This is based on the observation that only continental intraplate basalts show high Nb/Ta (up to 19) while MORB and OIB both have relatively low Nb/Ta (~15).

Reviewer #2 (Remarks to the Author):

The ms by Tang et al. "Building continents in magmatic orogens: Nb/Ta systematics in arc magma differentiation and the role of arclogites" presents and discusses data on Nb/Ta systematics in arc magmas and the implications for the mechanisms of continental crust formation.

I find the ms well written and the topic addressed is suitable for a broad audience like that of Nature Communications. The authors convincingly show that Nb/Ta systematics of arc magmas differ between those from thin and thick arc settings. Correlations between Nb/Ta and Dy/Yb are taken to indicate that the low Nb/Ta values of the thick continental crust are produced in concomitance with deep level fractionation of arc magmas (in the stability field of garnet) in the thick arc setting compared to the thin one. Through analyses of rutile (LA-ICPMS) and P-dependent solubility models of rutile they also show that the potential responsible for the low Nb/Ta of the continental crust is rutile in the garnet-pyroxene cumulates that are the product of the deep crustal fractionation.

Overall I find this ms very interesting for the understanding of continental crust formation mechanisms and that it can be accepted after very minor points are addressed, as outlined below and in the annotations of the attached file:

1. I would like to invite the authors to consider also the role of zircon and titanite in controlling the Dy/Yb ratio and therefore its correlation with Nb/Ta. This possibility may be especially important concerning high SiO₂ rocks for which fractionation of these two minerals is highly probable. The main line of reasoning would not change essentially because Dy/Yb would anyway be controlled by crystal fractionation.

We agree with the reviewer that zircon and titanite (sphene) are important fractionating phases as magmas differentiate towards more evolved compositions. Heavy rare earth elements are indeed compatible in both zircon and sphene. Sphene is also an important host of Nb and Ta during late stage differentiation. However, zircon probably has little effect on Dy/Yb fractionation because of its low abundance. The fact that magma Dy/Yb fractionation correlates with crustal thickness and thus pressure also indicates limited influence of zircon fractionation on magma Dy/Yb because pressure has no effect on zircon saturation in silicate melts (Boehnke et al., 2013).

While titanite is an important host of rare earth elements and high field strength elements, titanite prefers middle rare earth elements (such as Dy) over heavy rare earth elements (such as Yb) (Green and Pearson, 1987), and Ta over Nb (Green and Pearson, 1987; Tiepolo et al., 2002). Thereby, titanite fractionation should decrease Dy/Yb and increase Nb/Ta in the melts. We have now added some discussion about the role of titanite in the Methods section (line 571-575) and provided new data of Nb/Ta in titanite from Peninsular Ranges batholith in the supplement.

2. You should specify on which rutiles you performed LA-ICPMS analyses: are these the rutiles from the

cumulates or others? Is there any meaning for the large Nb/Ta range of values of these rutiles? i.e., dependence on setting, rock type, etc. in the case these rutiles are from different rocks and/or settings or also if they are all from cumulates? For this reason it is important to specify the petrogenetic context of the rutiles that you analysed.

Yes, these rutiles are from the cumulates studied here. We have now made this clear in the paper. The large variation in Nb/Ta in these rutiles is indeed interesting. There are several possibilities. First, kinetic fractionation between Nb and Ta during the growth of rutile may generate Nb/Ta variation in the rutile (Marschall et al., 2013). Second, exsolution of ilmenite from rutile during decompression and subsequent rutile-ilmenite re-equilibration may also produce Nb/Ta variation in rutile.

3. Lines 136-138 (now in Line 140-142) : “That rutile and Fe-Ti oxides are saturated as magmatic phases only in the low Mg# cumulates, suggests their crystallization from evolved and hence cooler magmas as the low Mg# cumulates are thought to be complementary to andesitic to dacitic magmas”: Perhaps it would be good to have some firmer evidence for this (i.e. modeling based on Mg# of cumulates versus andesite-dacite) or at least a reference if there is one.

We have now cited Tang et al. (2018) to support this argument. In Tang et al. (2018), we estimated cumulate-melt Mg# relationship by p-MELTS simulation. The melt in equilibrium with a cumulate of Mg#=0.6 has a Mg# of 0.3-0.4, and is moderately differentiated.

A few minor edits are reported in the attached file

Line 58. (now in Line 57) : You explain later what is the reason of the Nb/Ta decrease in thick arc settings. Do you have any idea of what is the reason for the (late) Nb/Ta decrease in thin arc settings?

The late Nb/Ta decrease in thin arc settings may also result from rutile fractionation. But in thin arcs, the pressure is low, and rutile can only come out at much lower temperatures.

We fixed the typos pointed out by the reviewer.

I think that this ms can be an influential work for the understanding of the mechanisms of continental crust formation and will be of interest for a broad community of the Earth Sciences.

Reviewer #3 (Remarks to the Author):

General remarks

The manuscript aims to provide a model to explain the overall low Nb/Ta in the continental crust by intracrustal Nb/Ta fractionation, where Ti-rich phases (rutile) play a major role. It is of great importance for the geochemistry community that deals with the differentiation of the solid Earth in a wider sense, and the outcome is original and novel and supported by the data. The manuscript is largely well and convincingly written and clearly organized, a few exceptions are marked in the PDF (towards the end, the text get's a bit more difficult to read and to follow, and a few slips of the pen need to be removed). The scientific idea is smart, the approach and concept are appropriate, and the results are significant for a better understanding of continental crust formation starting from an originally ultramafic source rock. With this regard, it might influence thinking in this field of research. The observation that on a very global scale MOR basalts and island arc basalts share

about identical Nb/Ta, and the derived conclusion that Nb/Ta differentiation to make low Nb/Ta continental crust therefore needs to be an intracrustal process is highly important. Based on this conclusion the manuscript, to my knowledge for the first time from this perspective, provides a very plausible mechanism to explain the overall accepted but rarely questioned low Nb/Ta in continental crust. With this respect it is a very important contribution to the "Nb paradox" discussion. The methodological approach is sound, the own data support the conclusions, and the data taken from the Georoc database have been filtered, and potential corruptions of Nb/Ta ratios were briefly discussed in the methods chapter. The scientific outcome is qualitatively supported by the data, but the manuscript misses a desirable quantitative support of the major conclusion (see comment to line 63-65 below).

Several weak points need to be also addressed, although they do not interfere the overall importance of the manuscript. So, the actual knowledge regarding the "Nb paradox" discussion is not described in sufficient detail, or presented in a too oversimplified way (see below), possibly to more strongly justify the need of the present work. This must be improved but will not detract the overall importance of the study. Then, the very global aspect addressed, i.e. transferring the process to the formation of continental crust in general, seems not to be fully justified. Clearly island arc and passive continental margin magmatism is important in building the continental crust, but given that the key process does obviously only occur in very thick active continental margins, the question arises to what extent this process can be regarded as the fundamental one that solely controls the Nb/Ta distribution in the continental crust. Possibly the authors should reduce their smart and well supported model to "some" active continental margin magmas with a (very) specific setting (i.e. very thick arc crust, what happens before and how is this thickness achieved?). Except, they can provide additional arguments that show that continental crust formation does only occur in very thick active continental margins. Beside the overall plausible explanation for low Nb/Ta in the continental crust, the authors describe in brief the fate of arcogites by suggesting foundering them back into the mantle. Although this makes sense, the expected high Nb/Ta in the order of 19 or more (see below) are not observed in any ocean basin derived basalt. This is evident from the data presented in the manuscript (Fig. 4b), as well as from the literature. This is conflicting and needs to be resolved.

We agree with the reviewer. To our knowledge, there are two problems associated with Nb/Ta. The first one is the sub-chondritic Nb/Ta ratio of the bulk silicate Earth, as suggested by previous authors e.g. Münker et al. (2003); the second problem is the Nb/Ta fractionation during the formation of continental crust. Our work deals with the second problem on continental crust formation instead of the subchondritic Nb/Ta of the bulk silicate Earth. The arcogites may solve the low Nb/Ta ratio of the continental crust, but may not provide a complimentary reservoir for the entire silicate Earth. Other processes may be required to solve the bulk silicate Earth problem. So, we don't think our observations are conflicting with our model. We have now added a paragraph to discuss the implications of our work for the subchondritic Nb/Ta of the bulk silicate Earth (line 216-220).

We agree that the tectonic settings of continent formation are still debated, but subduction zone is the only place where low Nb/Ta magmas, like that of the continental crust, have been observed. Our observations point to a strong link between continental arc magmatism/differentiation and the formation of continental crust.

Crustal thickening is driven by tectonic compression and magma addition. Other evidence for the role of synmagmatic crustal thickening comes from the high FeO content of the arcogites, which complements the low FeO calc-alkaline continental crust. The major Fe hosting phase is garnet, which is only stable at high pressure and high water contents, which in turn requires crustal thickening.

The cited literature is appropriate, but some rather fundamental and important papers are missing (e.g. Münker et al. 2003, Science, 301, 84-87; Weyer et al. 2003, EPSL, 205, 309-324; Jochum et al., 2000, Meteoritics & Planetary Sciences, 35, 229-235). These amongst others will help to complement the description of the "Nb-paradox".

Thanks for the suggestion, and we have now added these references in the paper.

In summary, the presented model is a plausible and novel solution for broadly accepted low Nb/Ta in the continental crust, as it ascribes the Nb/Ta differentiation to intracrustal processes. I.e. it moves the location of global Nb/Ta fractionation from mantle settings to crustal settings, which is a novel perspective that makes sense. The underlying process is plausibly derived from own and literature data. What remains unresolved or at least vague is the fate of the arclogites. Foundering them back into the mantle seems unlikely as ocean basalts are too low in Nb/Ta. However, the authors themselves provide a potential solution to this problem by linking the arclogites to continental intraplate magmatism, as continental basalts share the required high Nb/Ta. Strengthening this issue on the expense of the foundering hypothesis will result in an overall plausible model and a consistent manuscript.

I strongly recommend publication of the manuscript after having refined the above mentioned issues and after having carefully addressed the points listed below and in the PDF file.

Specific comments

Lines 21-23 (now in Line 22-24). This statement is not valid as it is too oversimplified, but it is an important prerequisite to justify the major conclusions of this manuscript. The "state-of-the art" regarding the "Nb paradox" should be outlined more comprehensively. Depleted mantle can be as low as 10 in Nb/Ta (Weyer et al. 2003, EPSL, 205, 309-324), MORB is around 14 but with distinct variations that result from partial melting of different sources (see also Weyer et al. 2003), OIBs are 15-16 (Pfänder et al., 2007, EPSL, 254, 158-172) and intraplate basalts are higher, between 15.0 - 19.1 (Pfänder et al., 2012, CMP, 2012, 232-251). The variation in Nb/Ta in silicate rocks is undoubtedly low, but resolvable, and this is due to partial melting, but primarily a result of different sources involved during melt formation. This needs to be figured out more clearly.

To be more precise, we rephrased the sentence as "The similar Nb/Ta ratios (~15) of mid-ocean ridge basalts, ocean island basalts and their derivative liquids indicate that Nb and Ta behave similarly during mantle melting and basaltic differentiation". We agree that there are some source effects on Nb/Ta ratio of oceanic basalts, and there are probably domains in the mantle with Nb/Ta anomalies, but the broadly consistent Nb/Ta ratio between MORB and OIB and constant Nb/Ta with respect to two orders of magnitude variation of Nb concentrations (Fig. R1) in oceanic basalts suggest that Nb/Ta fractionation during mantle melting is limited. We have now cited Weyer and Pfänder's papers in this sentence.

Fig. R1. Nb/Ta ratio as a function of Nb concentration in mid ocean ridge basalts (MORB, $n = 2,959$) and ocean island basalts (OIB, $n = 2,824$). Samples are binned by their Nb concentrations (bin size = 5 ppm) and plotted as means and two standard errors.

Lines 35-36 (now in Line 35-37). Due to mass balance considerations, this seems to be unlikely on a global scale. How can arclogites compensate the volume of the whole continental crust? Where should

these be? If such a statement is presented, it should be quantified by model calculations to give a realistic idea about the proportions needed to compensate the Nb deficit of the continental crust.

Yes, we quantified the mass of foundered arclogites in the paper (line 173-181). The minimum mass of the foundered rutile-bearing arclogites is 0.3-2.5 times that of the remaining continental crust. This is quite significant indeed, but we don't think it is impossible. Massive lower crustal recycling has also been suggested by a number of papers (e.g., Jagoutz and Schmidt, 2013; Plank, 2005; Tang et al., 2015).

Line 38-39 (now in Line 39-40). Clearly continent formation is linked to plate tectonics, this is trivial.

We agree with the reviewer that continent formation is linked to plate tectonics, but the mechanism remains elusive. Our paper approaches this problem by highlighting the role of synmagmatic crustal thickening, which primarily happens at convergent margins and thus requires plate tectonics.

Line 44-45 (now in Line 45-46). This needs references. Stolz et al. 1996 (Geology, 24, 587-590) reported elevated Nb/Ta in K-rich arc lavas (25 – 33).

We agree that arc lavas occasionally do show elevated Nb/Ta ratios, but we are not sure if high Nb/Ta arc lavas are relevant here.

Fig. 1a: The backbone of the manuscript are the low Nb/Ta in a little more than four “bins”, i.e. comparatively little data points averaged from a number of samples from Georoc. It needs to be resolved, which locations and settings these samples represent, are they representative to place a global scale conclusion, or are these locations “exotic” ones? In fact, only very few data points have Nb/Ta as low as continental crust, and if so, at higher or even much higher SiO₂. This rises the question whether this model is appropriate to derive a model for low Nb/Ta in continental crust in general. Rutile fractionation in thick arc crust seems to be undoubtedly an important process, the database is sound, the trends in Fig. 1 are clear and the overall conclusion is correct, but is it valid on a global scale or rather in some exceptional cases/settings where highly differentiated melts evolve in very thick active continental margin crust? What is the role of amphibole fractionation in forming continental crust with low Nb/Ta as suggested by Li et al. (2017) (Journal Petrology, as cited in the manuscript)?

The number of bins is determined by the total range and bin size. Behind these limited bins with low Nb/Ta are 645 samples (SiO₂ = 65-75 wt.%) from thick continental arcs. In Fig. R2, we plot the histogram for differentiated rocks (SiO₂ = 65-75 wt.%) from thick continental arcs vs. island arcs and incipient continental arcs. We think the data is sufficient and the difference is significant. We have now added a panel similar to Fig. R2 in Fig. 1 in the manuscript.

Thick continental arc samples are from central and northern Andean arcs. These are the only places with > 50 km average crustal thickness and active arc magmatism. We have now added this information in the caption of Fig. 1.

Amphibole fractionation is potentially important in fractionating Nb/Ta in hydrous arc magma differentiation. But unlike rutile, amphibole has a wide stability field (Green, 1982) and its fractionation should happen in most arc systems, which makes it difficult to explain the systematic difference in Nb/Ta fractionation between thick arcs vs. thin and normal arcs. We thus argue that the effect of amphibole on Nb/Ta fractionation is likely limited due to the generally incompatible behavior of Nb and Ta in amphibole (partition coefficients mostly below 1).

Fig. R2. Histogram showing Nb/Ta ratio of differentiated arc magmas from thick mature continental arcs (crustal thickness > 50 km) vs. island arcs and incipient continental arcs (crustal thickness < 50 km)

Line 63-68 (now in Line 77-82), Caption Fig. 1. The volumes of highly differentiated arc magmas and arc basalts will be very different in continental arcs, with arc basalts dominating. I'd claim that to produce crust with Nb/Ta ~11 and SiO₂ ~62wt%, unrealistically high volumes/masses of differentiated magmas are required. Please provide calculated mixing lines and denote the portion of differentiated melt added on the tick marks. Also mark the end-member compositions with symbols in Fig. 1a.

Arc basalts are indeed the dominating lithologies in island arcs where the crust is thin. In continental arcs where the crust is thicker, andesitic and dacitic magmas dominate. Actually, the average crustal SiO₂ content increases with crustal thickness (Dhuime et al., 2015; Farner and Lee, 2017). Basalts are very rare in central and northern Andean arc, the thickest active arcs in the world today. The continental crust and continental arcs are composed of a series of lithologies spanning from basaltic to felsic. We considered two endmember mixing scenarios between basalt (SiO₂ = 50 wt.%) and andesite (SiO₂ = 60 wt.%) and between basalt and granite (SiO₂ = 70 wt.%). These two endmember mixing scenarios can bracket most mixing possibilities in terms of compositions. The continental crust Nb/Ta can be produced by mixing basaltic and felsic endmembers from continental arcs in the ratio of 1:1 to 1:3. These numbers appear to be realistic given the observed lithologies in modern continental arcs. We have now tick-marked the endmember mixing curves for both island arc and continental arc in Fig. 1a.

Line 116 (now in Line 122). ... until after ...

We have now rephrased the sentence as "rutile and/or Fe-Ti oxides saturate in the magma only after sufficient differentiation".

Line 117-118 (now in Line 122-124). This is not only valid for the low Mg# cumulate samples, but also for high Mg#, although their average might be lower? Is there really a difference?

The low Mg# cumulates have higher average Nb/Ta because of rutile. The high Mg# cumulates have an average basalt like, low Nb/Ta ratio and low Nb, Ta and Ti contents because they have no rutile.

Line 121-123 (now in Line 126-129). Provide an average Nb/Ta for the high Mg# samples, and 1s. What is the concentration weighed total cumulate average Nb/Ta value? Please denote.

The average Nb/Ta for the high Mg# cumulates is 12.6 ± 5.8 (1 σ), and concentration weighted mean Nb/Ta for the high Mg# cumulates is 15.0. We have now denoted the average Nb/Ta ratio for the high Mg# cumulates in the paper. However, We don't think these rutile-free high Mg# cumulates are important

because their Nb and Ta concentrations are 1-2 orders of magnitude lower than the low Mg#, rutile-bearing cumulates.

Line 124ff (now in Line 130). It is known since long that if present, Ti-bearing phases, in particular rutile, control the HFSE budget of a system. Needs no in depth evaluation, possibly cite one or two papers that provide consistent partition coefficients for Nb and Ta between rutile, Fe-Ti oxides and mafic magmas. Rudnick et al. (2000) (Science, 287, 278-281), and later Aulbach et al. (2008) (Nature Geosciences, 1, 468-472) both identified rutile as dominant host for Nb and Ta in subducted eclogites and in the subcontinental lithospheric mantle.

We agree that rutile has been demonstrated to be an important host of HFSEs, but it is debated whether rutile prefers Nb over Ta. We have now rephrased this sentence to be more accurate.

Rutiles from subducted slab appear to have high Nb/Ta (Rudnick et al., 2000), but high Nb/Ta in rutile had not been reproduced by experimental studies until Xiong et al. (2011), who showed that Nb/Ta partition coefficient ratio between rutile and melt is strongly dependent on temperature. Arc cumulates and subducted slabs can be different. So, we analyzed the cumulate rutiles to test if they have high Nb/Ta ratios.

Line 132-134 (now in Line 138-140). According to the equation given in the cited paper (Xiong et al., 2011), $D_{Nb/Ta}$ becomes >1 only in exceptional cases, i.e. for low water contents ($<4-5\text{wt}\%$), at relatively low temperatures ($<1000^\circ\text{C}$) and high pressures ($>2\text{ GPa}$). This results in very low geothermal gradients ($<15^\circ\text{C}/\text{km}$), to my opinion unrealistic in active continental margins with a long lasting magmatic history. Therefore, the provided explanation for the discrepancy between experimental data and observation is unsatisfying. The authors should test and more comprehensively discuss what the causes of this contradiction could be. It seems also unrealistic to assume such low water contents in highly differentiated arc magmas.

According to Xiong et al. (2011), there appears to be no pressure dependence on Nb/Ta partitioning between rutile and melt. So, the thermal gradient shouldn't matter here. Xiong et al. (2011) also concluded that D_{Nb}/D_{Ta} can exceed 1 when the temperature is below 1000°C and melt water content below $10\text{ wt.}\%$. These conditions appear to be realistic for andesitic to dacitic magmas.

Lines 168-171 (now in Line 174-177). Unclear. Please explain properly and comprehensively what you did and what the assumptions are.

We have now added a new subsection (3.1) in the Methods to explain our mass balance calculation in more details. We hope this is clearer now.

Line 171-176 (now in Line 177-181). A foundering model is favored to remove the garnet-pyroxenite cumulates with high Nb/Ta from the base of the thickened crust. The mass estimate based on averaged concentrations seems to yield a mass of foundered material of at least 2.5 times the mass of the continental crust. With this respect, this model resembles the eclogite recycling model of Rudnick et al. (2000) (see above). Given that the recycled material of either subducted or foundered garnet-pyroxenite ("eclogite") has a lower solidus temperature than ambient mantle peridotite (as outlined in lines 188-190), partial melts of such sources should be at least partly sampled by oceanic magmatism such as OIBs. Those should therefore indicate elevated Nb/Ta, i.e. >18.8 if the averaged cumulate Nb/Ta of this study is taken as source value, as partial melting, even in the presence of rutile, will further enlarge Nb/Ta in the melt relative to the source. This, however, is not the case as is visible from Figure 4b (for high precision Nb/Ta in OIBs and melting modelling see also Pfänder et al., 2007, EPSL, 254, 158-172). Any other option of storage? Is the foundering necessarily required?

The garnet-pyroxenite arc lower crust is gravitationally unstable, and should be eventually removed. Why OIBs do not see signals of recycled arclogites is indeed interesting. It is possible that these arclogites

may not be efficiently recycled to the source of OIBs, or the rutile may be completely exhausted during early melting. Future studies may help to clarify this.

Line 199-200 (now in Line 203-205). I would engage the authors to undertake melting calculations to verify the qualitative statements made. As outlined before, Nb/Ta in melts from foundered garnet-pyroxenite with Nb/Ta~18-19 should be at least equal or higher, which is not observed taking the sample compilation in Fig. 4b and literature data (Pfänder et al., 2007).

See above. We think that arclites, or at least the rutile in the arclites, may not be efficiently recycled to the source of OIBs.

Line 200 (now in Line 205). The cited reference (Pfänder et al., 2012) invokes carbonatitic melts to explain elevated Nb/Ta in continental basalts, not pyroxenite melts, which, however, might be a good alternative based on the observations made here and as discussed in lines 202-206.

Fixed.

Line 467-471 (now in Line 551-553, Extended Data Fig. 4). Extended Data Figure 3b displays that at overall low Ta contents the solution based measurements provide significantly lower concentrations than the LA-ICP-MS measurements. The explanation for this is weak. Explaining it by low count rates for LA-ICP-MS is unsatisfying. This should result in the opposite. Either this is background or an isobaric interference during LA-ICP-MS, or the solution method loses Ta, possibly by adsorption to teflon vial walls, why Münker et al. (2001) (G-cubed, Vol. 2, 10.1029/2001GC000183) have suggested to use traces HF in all acid steps during HFSE analyses in solution based methods. It is crucial to know which of the methods is appropriate.

We have now added a new figure (Extended Data Fig. 3) in the Methods to compare our solution ICP-MS results with GeoRem recommended values for six standards (BCR-2, BIR-1, BHVO-2, AGV-2, GSP-2 and RGM-2). They all agree within 3%. So, we think our solution ICP-MS method is robust and Ta loss is highly unlikely. Ta loss would also make it difficult to explain the consistency between the solution ICP-MS data and LA-ICP-MS data for high concentration samples. The discrepancy between the solution and LA data probably arises from the LA measurements when determining Ta concentrations < 0.05 ppm. This is close to the detection limit of LA-ICP-MS, considering that Ta in the glass discs is further diluted by lithium metaborate, the base material of the glass.

Lines 488-489 (now in Line 567-568). Without being an expert in LA-ICP-MS, I'd expect that a silicate glass standard will release and fractionate different elements in a different manner than oxides, with different bond strengths and crystal lattice geometries. Are there no oxide standards (rutile) available? Calibrated natural rutile or Fe-Ti oxides to minimize matrix effects?

To our knowledge, there are no rutile standards with homogenous Nb and Ta compositions in the market. However, Luvizotto et al. (2009) showed that LA-ICP-MS can yield accurate results for Nb and Ta in rutile using NIST glass standards for calibration.

Line 498 (now in Line 595). Are errors of the coefficients of the equation given in the original publication?

No, the authors did not provide the errors associated with the coefficients.

Line 504 (now in Line 602). This SiO₂ holds for erupted arc magmas (from which the curve in Extended Data Figure 4 is derived), but prior to fractionation and during fractionation, it should be lower. Is there a lower SiO₂-limit in the fractionating magma where the pressure needs to be unrealistically high to become rutile to be a liquidus phase?

We did the rutile saturation calculation for this SiO₂ range because the island arc and continental arc Nb/Ta-SiO₂ trends diverge at SiO₂ = 60–65 wt.%, suggesting the onset of rutile saturation in continental

arc magmas at $\text{SiO}_2 = 60\text{--}65$ wt.%. The melt composition parameter FM is in the range of 3-4 at $\text{SiO}_2 = 60\text{--}65$ wt.%. Using an FM of 3, the pressure to saturate rutile in andesitic to dacitic magmas falls in the range of 1.2-1.5 GPa (40-50 km), as discussed in the paper. We think this is a reasonable pressure range for magma differentiation in thick continental arcs.

Extended Data 1: Would suggest to add the Nb/Ta ratios of the standards and compare them to the ratios calculated from the recommended values. An additional comparison can be made to Pfänder et al. (2007; see above), who provides Ta-ID concentration values for BHVO-2 and high-precision Nb/Ta for this standard.

Done.

Comments in the PDF

Line 16 (now in Line 17): True? By volume, or by area, or by mass? CC is thicker, about two to three fold than oceanic crust, so this may balance the lower surface area of CC with respect to mass (volume).

We agree. We have now rephrased the sentence.

Line 91 (now in Line 95): in differentiated lithologies

Fixed.

Line 121 (now in Line 126): Mark the divide between low and high Mg# samples in Fig. 2b.

We put grey shades in all panels of Fig. 2 to indicate the oxide present field. So the oxide-in line marks the divide between high and low Mg# samples. We have also made it clear at the beginning of this paragraph that low Mg# cumulate samples are those with $\text{Mg\#} < 0.6$.

Line 132 (now in Line 138): what melt?

The melts in equilibrium with rutile in the experiments.

Line 161 (now in Line 167): Two times depth... redundant

Fixed.

Line 168 (now in Line 173): a mass balance is not something to assume, a mass balance is something that is calculated

We have now rephrased this sentence as "We can estimate the mass of foundered rutile-bearing arclogites using Nb and Ta mass balance".

Line 404 (now in Line 471): by

Fixed.

Line 431 (now in Line 502): Not easy to get, this sentence seems to be out of context, should be improved.

There are two ways to calculate ratios. First, one can calculate the Nb/Ta ratio for each sample, and then average the Nb/Ta ratios of all samples. Or, one can calculate the average Nb and Ta concentrations for all samples, and then calculate the concentration weighted mean Nb/Ta using the average Nb and Ta concentrations. We did both, and we would like to make it clear how we calculated the ratios here.

Line 469-471 (now in Line 547-550): weak argument

See reply above regarding the discrepancy between the solution ICP-MS and LA-ICPMS data for low Ta samples.

References

- Alonso-Perez, R., Müntener, O., Ulmer, P., 2009. Igneous garnet and amphibole fractionation in the roots of island arcs: experimental constraints on andesitic liquids. *Contributions to Mineralogy and Petrology* 157, 541-558.
- Boehnke, P., Watson, E.B., Trail, D., Harrison, T.M., Schmitt, A.K., 2013. Zircon saturation re-revisited. *Chemical Geology* 351, 324-334.
- Dhuime, B., Wuestefeld, A., Hawkesworth, C.J., 2015. Emergence of modern continental crust about 3 billion years ago. *Nature Geosci* 8, 552-555.
- Farner, M.J., Lee, C.-T.A., 2017. Effects of crustal thickness on magmatic differentiation in subduction zone volcanism: A global study. *Earth and Planetary Science Letters* 470, 96-107.
- Green, T.H., 1972. Crystallization of calc-alkaline andesite under controlled high-pressure hydrous conditions. *Contributions to Mineralogy and Petrology* 34, 150-166.
- Green, T.H., 1982. Anatexis of mafic crust and high pressure crystallization of andesite, in: Thorpe, R.S. (Ed.), *Andesites: orogenic andesites and related rocks*. John Wiley & Sons, New York, pp. 465–487.
- Green, T.H., Pearson, N.J., 1987. An experimental study of Nb and Ta partitioning between Ti-rich minerals and silicate liquids at high pressure and temperature. *Geochimica et Cosmochimica Acta* 51, 55-62.
- Jagoutz, O., Schmidt, M.W., 2013. The composition of the foundered complement to the continental crust and a re-evaluation of fluxes in arcs. *Earth and Planetary Science Letters* 371–372, 177-190.
- Kerr, A.C., 2003. Oceanic plateaus. *Treatise on geochemistry* 3, 659.
- Luvizotto, G.L., Zack, T., Meyer, H.P., Ludwig, T., Triebold, S., Kronz, A., Münker, C., Stockli, D.F., Prowatke, S., Klemme, S., Jacob, D.E., von Eynatten, H., 2009. Rutile crystals as potential trace element and isotope mineral standards for microanalysis. *Chemical Geology* 261, 346-369.
- Marschall, H.R., Dohmen, R., Ludwig, T., 2013. Diffusion-induced fractionation of niobium and tantalum during continental crust formation. *Earth and Planetary Science Letters* 375, 361-371.
- Münker, C., Pfänder, J.A., Weyer, S., Büchl, A., Kleine, T., Mezger, K., 2003. Evolution of planetary cores and the Earth-Moon system from Nb/Ta systematics. *Science* 301, 84-87.
- Plank, T., 2005. Constraints from Thorium/Lanthanum on Sediment Recycling at Subduction Zones and the Evolution of the Continents. *Journal of Petrology* 46, 921-944.
- Rudnick, R.L., Barth, M., Horn, I., McDonough, W.F., 2000. Rutile-Bearing Refractory Eclogites: Missing Link Between Continents and Depleted Mantle. *Science* 287, 278-281.
- Tang, M., Erdman, M., Eldridge, G., Lee, C.-T.A., 2018. The redox “filter” beneath magmatic orogens and the formation of continental crust. *Science Advances* 4.

Tang, M., Rudnick, R.L., McDonough, W.F., Gaschnig, R.M., Huang, Y., 2015. Europium anomalies constrain the mass of recycled lower continental crust. *Geology* 43, 703-706.

Tiepolo, M., Oberti, R., Vannucci, R., 2002. Trace-element incorporation in titanite: constraints from experimentally determined solid/liquid partition coefficients. *Chemical Geology* 191, 105-119.

Xiong, X., Keppler, H., Audétat, A., Ni, H., Sun, W., Li, Y., 2011. Partitioning of Nb and Ta between rutile and felsic melt and the fractionation of Nb/Ta during partial melting of hydrous metabasalt. *Geochimica et Cosmochimica Acta* 75, 1673-1692.

REVIEWERS' COMMENTS:

Reviewer #3 (Remarks to the Author):

Dear Authors, dear Editors,

having checked the rebuttal letter of the authors, and the revised manuscript, I find that the revisions might be a bit scarce at one or another point, but they are overall acceptable and mostly consider the thoughts contributed by the reviewers. Nevertheless, two difficult issues remain, and these are (1) the low geothermal gradient that seems to be necessary to achieve $D(\text{Nb}/\text{Ta}) > 1$ in rutile, and (2) the standardisation of the LA-ICPMS method by using standard glasses to measure rutile compositions. I see that these two points are not easily to overcome, but the imagination that there might be a small but realistic likelihood that high Nb/Ta in rutiles are an artefact due to a standardization problem is seriously a nightmare. I therefore would like to see the problem with the unexpectedly low geothermal gradient to achieve $D(\text{Nb}/\text{Ta})$ in rutile > 1 mentioned and discussed in the paper, in the context of the discussion of the obvious discrepancy between high Nb/Ta in the rutiles but overall $D(\text{Nb}/\text{Ta}) < 1$ in most experiments. And second, I would like to see a short hint on possible problems that might arise from using glass standards to calibrate LA-ICPMS protocols for rutile/oxide measurements.

Having considered these final issues and some other very minor points as annotated in the attached documents (manuscript text and rebuttal text), the manuscript is ready for publication in Nature Communications, and I'm looking forward seeing it online. It is doubtless an important piece of science.

Revision notes

We thank again Reviewer #3 for the comments. We have made the required changes accordingly. We have also shortened the abstract to 157 words. The paper now has Introduction, Results, Discussion and implications sections. In the paper, we marked revised texts in red. Below is our detailed response to Reviewer #3's comments.

Reviewer #3 (Remarks to the Author):

Dear Authors, dear Editors,

having checked the rebuttal letter of the authors, and the revised manuscript, I find that the revisions might be a bit scarce at one or another point, but they are overall acceptable and mostly consider the thoughts contributed by the reviewers. Nevertheless, two difficult issues remain, and these are (1) the low geothermal gradient that seems to be necessary to achieve $D(\text{Nb}/\text{Ta}) > 1$ in rutile, and (2) the standardisation of the LA-ICPMS method by using standard glasses to measure rutile compositions. I see that these two points are not easily to overcome, but the imagination that there might be a small but realistic likelihood that high Nb/Ta in rutiles are an artefact due to a standardization problem is seriously a nightmare. I therefore would like to see the problem with the unexpectedly low geothermal gradient to achieve $D(\text{Nb}/\text{Ta})$ in rutile > 1 mentioned and discussed in the paper, in the context of the discussion of the obvious discrepancy between high Nb/Ta in the rutiles but overall $D(\text{Nb}/\text{Ta}) < 1$ in most experiments.

We agree. The high $D_{\text{Nb}}/D_{\text{Ta}}$ between rutile and melt requires crystallization from low temperature evolved melt, which may not translate into low geothermal gradient. This is because melt temperature depends mostly on composition and pressure.

The high $D_{\text{Nb}}/D_{\text{Ta}}$ between rutile and melt is inconsistent with early experiments but agrees with the latest experiment work by Xiong et al. (2011, GCA). At temperature < 1000 C and water content $< 10\%$, $D_{\text{Nb}}/D_{\text{Ta}}$ between rutile and melt exceeds 1, as pointed out by Xiong and co-workers. We have now emphasized this point in line 135-144.

And second, I would like to see a short hint on possible problems that might arise from using glass standards to calibrate LA-ICPMS protocols for rutile/oxide measurements.

We have added a few lines to note this potential issue (line 539-542). However, we found that matrix dependent fractionation is very limited in LA-ICP-MS measurements of oxide Nb and Ta, as shown by the experimental work by Xiong et al. (2011, GCA) and the rutile standard characterization work by Luvizotto et al. (2009, Chem. Geol.).

Having considered these final issues and some other very minor points as annotated in the attached documents (manuscript text and rebuttal text), the manuscript is ready for publication

in Nature Communications, and I'm looking forward seeing it online. It is doubtless an important piece of science.